# Combining formal methods and Bayesian approach for inferring discrete-state stochastic models from steady-state data

**Julia Klein**[1,2], **Huy Phung**[1], **Matej Hajnal**[1,2,3], **David Šafránek**[3], **Tatjana Petrov**[1,2]*

**1** Department of Computer and Information Sciences, University of Konstanz, Konstanz, Germany, **2** Centre for the Advanced Study of Collective Behaviour, University of Konstanz, Konstanz, Germany, **3** Systems Biology Laboratory, Faculty of Informatics, Masaryk University, Brno, Czech Republic

* tatjana.petrov@gmail.com

**Data Availability Statement:** Data are available from https://zenodo.org/record/7900258#. ZFzblexBwqs.

## Abstract

Stochastic population models are widely used to model phenomena in different areas such as cyber-physical systems, chemical kinetics, collective animal behaviour, and beyond. Quantitative analysis of stochastic population models easily becomes challenging due to the combinatorial number of possible states of the population. Moreover, while the modeller easily hypothesises the mechanistic aspects of the model, the quantitative parameters associated to these mechanistic transitions are difficult or impossible to measure directly. In this paper, we investigate how formal verification methods can aid parameter inference for population discrete-time Markov chains in a scenario where only a limited sample of population-level data measurements—sample distributions among terminal states—are available. We first discuss the parameter identifiability and uncertainty quantification in this setup, as well as how the existing techniques of formal parameter synthesis and Bayesian inference apply. Then, we propose and implement four different methods, three of which incorporate formal parameter synthesis as a pre-computation step. We empirically evaluate the performance of the proposed methods over four representative case studies. We find that our proposed methods incorporating formal parameter synthesis as a pre-computation step allow us to significantly enhance the accuracy, precision, and scalability of inference. Specifically, in the case of unidentifiable parameters, we accurately capture the subspace of parameters which is data-compliant at a desired confidence level.

## Introduction

Population models are widely used to model different phenomena: animal collectives such as social insects, flocking birds, schooling fish, or humans within societies, as well as molecular species inside a cell, or cells forming a tissue. Quantitative models of the underlying mechanisms can directly serve important societal actions such as disaster response (for example, mitigating the spread of epidemics [1]), they can inspire the design of distributed algorithms (for example, ant colony algorithm [2]), or aid robust design and engineering of collective, adaptive

**Funding:** This study was supported in the form of funding by Grantová Agentura České Republiky, (Grant No. GA22-10845S) awarded to DS, by DFG Centre of Excellence 2117 'Centre for the Advanced Study of Collective Behaviour' (ID: 422037984) awarded to TP, JK, MH, and HP, by the Ministry of Science, Research and the Arts of the state of Baden-Württemberg awarded to TP, by the Young Scholar Fund (YSF), (project no. P83943018FP430_/18) awarded to TP, and by AFF (Der Ausschuss für Forschungsfragen, EU-Anschubfinanzierung, Univ. of Konstanz) awarded to JK. The funders had no role in study design, data collection and analysis, decision to publish, or preparation of the manuscript.

**Competing interests:** The authors have declared that no competing interests exist.

systems under given functionality and resources, which is recently gaining attention in vision of smart cities [3, 4]. In practice, the qualitative aspects of population models—the existence of connections between different population states—are usually easy to hypothesise, as they can be inferred from the local interaction mechanisms between individual agents. However, precise analysis of the population model as a whole necessitates corresponding quantitative explanations. To this end, computational modelling with population models easily becomes challenging, because the model parameters are often uncertain or unknown, and measuring them experimentally is difficult or impossible. At the same time, the available experimental data typically measures aggregate, population-level quantities at chosen time instances or only after the system's dynamics has stabilised to a stationary regime [5].

In this paper, we tackle the problem of parameter inference for a wide class of stochastic population models called discrete-time Markov chains (DTMCs), in a common scenario where the measurements are made only once the system's dynamics have stabilised to a stationary regime.Such steady-state experimental data are a wide-spread format of data in biology for several reasons. First, the states reached after the system entered a stationary regime, often called *terminal states* (or, more generally, set of terminal states or *terminal components*), do not change after they are once reached; they can be reliably observed at any point of time after the transient phase, hence avoiding uncertainties due to event delays and event synchronisation during the transient. It is worth mentioning that the methodology we propose in this paper can be applied only if there exist multiple, different terminal components. Yet, numerous phenomena relevant in practice exhibit such feature. For instance, in biological systems, multiple terminal observations are pervasive, as they implement phenotypic diversity (think of, for example, genetic switches in cell differentiation [6, 7], or multi-stability phenomena observed in the immune response [8, 9], or cell cycle control [10, 11]). Moreover, beyond the application context of modelling biological systems, different terminal states typically encode different outputs in randomised algorithms (for instance, a typical example is a randomised algorithm encoding a six-sided die, where each of the six outcomes is encoded through a different terminal state).

Here, we concretely consider a DTMC with multiple terminal states, with known structure and a finite set of unknown parameters which influence the transition probabilities. More precisely, the formal object we work with is called a parametric Markov chain (pMC), in which the transition probabilities of the chain can be any rational function over a finite set of unknown parameters. Then, we assume a situation where we can make repeated data observations which of the multiple terminal states was reached after an extended period of time. Our goal is to find the space of parameters of the chain which is viable with respect to such data. In other words, we consider parameter inference for pMC, when only a sample distribution of reaching different terminal states of the chain is available.

Parameter inference in the case of steady-state data is challenging because the data may not provide enough information to identify the parameters of the chain [12]. We show that, if there exist multiple, different terminal components, the variability of terminal outcomes may suffice to identify some or all system parameters. Yet, even when all parameters are identifiable, inference involves various sources of uncertainty [13]. First, there is uncertainty due to a limited sample size. Moreover, the likelihood function for steady-state observations in a parametric Markov chain is typically not available in its analytical form and has to be approximated. Furthermore, the standard sampling-based Bayesian inference approaches involve additional uncertainty with respect to the choice of prior distributions, number of perturbation kernels, particles, and simulation length. As a result, while these traditional algorithms often give informative results within an available time frame, optimising their performance is difficult.

To address these challenges, we propose to use *formal methods for parameter synthesis* to aid parameter inference. Formal methods employ a variety of theoretical computer science fundamentals and were originally developed for the design, analysis and verification of software systems. Today, they also serve as a technique for the mathematically rigorous modelling of, for example, cyber-physical and biological systems [14, 15]. We first employ formal methods to obtain the exact likelihood for given data in terms of rational functions over parameters of the MC: we recast the data observations into a set of temporal properties and leverage the parameter synthesis tools to obtain the rational functions that exactly characterise reachability of respective terminal states (components). Then, we implement methods employing these rational functions to:

- compute the viable space of parameters of the chain compliant with data in a traditional frequentist interpretation of uncertainty,

- reduce uncertainty and boost scalability in an MCMC parameter inference scheme,

- efficiently infer parameter points closest to data observations (which is applicable only in the case of identifiable parameters).

Results demonstrate that pre-computing the rational functions with formal methods allows us to, in case of identifiable parameters, significantly enhance the accuracy, precision, and scalability of inference with respect to the sampling-based, likelihood-free technique. Moreover, in case of unidentifiable parameters, where the traditional techniques infer unreliable single-point estimates from the available data, our method accurately captures the viable subspace of parameters which are data-compliant at a desired confidence level.

The paper is organised as follows. In Section Preliminaries, we define preliminaries and discuss the parameter identifiability and uncertainty over a motivating example. In Section Methods, we propose how to compute exact likelihoods in the form of rational functions by encoding the terminal states (components) as a reachability property and leveraging the general-purpose tools for probabilistic model checking [16, 17]. Once symbolic forms of likelihood functions are obtained, we propose three generic algorithms for inferring parameters of pMC from steady-state data. Moreover, for the purpose of comparison, we introduce a likelihood-free Bayesian inference algorithm combining sequential Monte Carlo and approximate Bayesian computation idea s (SMC-ABC). In Section Case Studies, we report results over four case studies: an artificial nested branching model, a honeybee mass-stinging study, the SIR model, and the Zeroconf protocol.

## Related works

There exists a substantial body of work on verification of discrete-time pMC, subject to temporal logic properties: symbolic computation of reachability properties through state elimination in a parametric Markov chain [18, 19], lifting the parameters towards verifying a non-parametric Markov decision process (MDP) instead of the original pMC [20], candidate region generation and checking, helped by satisfiability modulo theories (SMT) solvers (see [21] and references therein; SMT solvers are tools to determine whether a logical first-order formula is satisfiable. Based on SAT solvers, SMT solvers were developed to support decision problems with respect to different background theories). Specifying biological properties as temporal logic formulae, and using such specifications for parameter synthesis, has already been applied in biological modelling: in [22], the authors compute the robustness of evolving gene regulatory networks by first synthesising the viable space of parameters with constraint solvers. In a related setup in [23–25], the authors express properties of general biochemical reaction

networks in continuous signalling logic (CSL), where they deal with the parameter synthesis for continuous-time Markov chains. Recently, in [26], direct integration of data into Bayesian verification of parametric chains has been proposed, designed to handle affine transition functions in the pMC, while in [27], the authors propose a grey-box model-checking framework for continuous-time chains (CTMCs), using likelihood-free parameter inference. In [5], the authors study a bridging problem (inference under terminal constraints) for CTMCs. To the best of our knowledge, the latter framework could not directly handle our case study, because it is designed to handle affine transition functions in the pMC. The idea of encoding data observations into reachability properties to obtain likelihood functions and subsequently apply parameter synthesis with SMT solvers was first introduced in our preliminary works in [28] and further elaborated towards shedding light on how honeybees adapt their defence in social context [29]. We here extend our original idea by elucidating the parameter identifiability and uncertainty propagation when only steady-state data are available, and investigating how obtained rational functions can be coupled with Bayesian inference. In our computational experiments, we considered using several tools which support parameter synthesis—PRISM [16], Prophesy [30], and Storm [17]. Finally, we used PRISM as it supports a command line input, helpful for the automatisation of the workflow.

## Preliminaries

In this section, we briefly introduce the formal objects used throughout this paper. The set of real numbers will be denoted by $\mathbb{R}$.

**Definition 1 (MC)** A Markov Chain is a tuple $\mathcal{M} = (S, P, \iota_{init}, AP, L)$ over a countable, nonempty set of states $S$, the transition probability function $P : S \times S \to [0, 1]$ such that $\Sigma_{s' \in S} P(s, s') = 1$ for all $s \in S$, the initial distribution $\iota_{init} : S \to [0, 1]$ such that $\Sigma_{s \in S} \iota_{init}(s) = 1$, a set of atomic propositions $AP$, and a state-labelling function $L : S \to 2^{AP}$.

Given an MC $\mathcal{M} = (S, P, \iota_{init}, AP, L)$, the probability space is assigned in the standard way, i.e. for any $l \geq 0$, the prefix set of traces $\sigma = (s_0, s_1, \ldots, s_l) \in S^{l+1}$ is assigned the probability measure $\mathsf{P}_{\mathcal{M}}(\sigma) = \iota_{init}(s_0) \prod_{i=0}^{l-1} P(s_i, s_{i+1})$. The property of reaching a terminal state (component) in a Markov Chain can be written in the temporal logic PCTL (Probabilistic Computational Tree Logic) [31]. We here consider a fragment of PCTL properties for persistence (FG) properties. These are defined over the traces for MC $\mathcal{M}$ in a standard way by state formulae induced by the grammar $\Phi ::= \text{true} \mid a \mid \Phi \mid \Phi_1 \wedge \Phi_2 \mid \Phi_1 \vee \Phi_2 \mid P_J(\phi)$, where $a \in AP$, $\phi$ is a path formula, and $J \subseteq [0, 1]$ is an interval, and path formulae $\phi ::= \Phi_1 \mathsf{U} \Phi_2$, where $\Phi_1, \Phi_2$ are state formulas, and $\mathsf{U}$ is the usual interpretation of an Until operator. We will write $\mathsf{P}(\mathcal{M} \models \phi) = \sum_{s \in S} \iota_{init}(s) \mathsf{P}_{\mathcal{M}}(s \models \phi)$ to denote the probability of satisfaction of PCTL property $\phi$ in the MC $\mathcal{M}$.

When the transition probabilities are not known, but rather are rational functions of some parameters from the parameter set $\mathcal{V}$, each over domain $[0, 1]$, we work with a parametric Markov Chain (pMC). We here restrict our attention to the case when the transition probabilities are multivariate rational functions over the variables $\mathcal{V}$, which we will denote by $\mathsf{Rat}_{\mathcal{V}}$. In general, the reachability probabilities for a pMC can be expressed by rational functions; in case studies shown in this paper, polynomials will suffice.

**Definition 2 (pMC)** A Parametric Markov Chain (pMC) is a tuple $\mathcal{M}_{\mathcal{V}} = (S, P_{\mathcal{V}}, \iota_{init}, AP, L, \mathcal{V})$, where $P_{\mathcal{V}} : S \times S \to \mathsf{Rat}_{\mathcal{V}}$ defines the probability transition matrix, and for each evaluation of parameters $\theta \in [0, 1]^{|\mathcal{V}|}$ induces a well-defined Markov chain $\mathcal{M}(\theta) = (S, P_{\theta}, \iota_{init}, AP, L)$, where $P_{\theta}$ denotes the instantiation of the expression $P_{\mathcal{V}}$, for parameter evaluations given by a vector $\theta$. Consequently, for any $\theta \in [0, 1]^{|\mathcal{V}|}$, for all $s \in S$, $\Sigma_{s' \in S} P_{\theta}(s, s') = 1$.

What we have previously referred to as terminal states (components) will now be formally described and replaced by the term bottom strongly connected component (BSCC). Both terms will be used interchangeably in the remainder of the paper.

**Definition 3 (BSCC)** A subset $T$ of $S$ is called strongly connected if for each pair $(s, t)$ of states in $T$ there exists a path fragment $s_0 s_1 \ldots s_n$ such that $s_i \in T$ for $0 \leq i \leq n$, $s_0 = s$ and $s_n = t$, and $P(s_i, s_{i+1}) > 0$ for all $i = 0, \ldots, n - 1$. A strongly connected component (SCC) of $\mathcal{M}$ denotes a strongly connected set of states such that no proper superset of $T$ is strongly connected. A bottom SCC (BSCC) of $\mathcal{M}$ is an SCC $T$ from which no state outside $T$ is reachable, i.e. for each state $t \in T$ it holds that $P(t, T) = 1$.

We denote the steady-state distribution of a MC by $\mu : S \to [0, 1]$ and the steady-state probability for a single state $s \in S$ by $\mu_s$. Since almost surely (with probability 1) any finite Markov Chain eventually reaches a BSCC and visits all its states infinitely often, the steady-state distributes the probability mass among its BSCCs, i.e. $\sum_{T \in BSCC(\mathcal{M})} \sum_{s \in T} \mu_s = 1$ [32].

We will use Bayesian approaches to estimate parameters agreeing with data.

**Definition 4 (Bayes theorem)** Let $\pi(\theta)$ denote the prior distribution over parameter(s), $P(D_{obs}|\theta)$ the likelihood of data observations under given parameters, and $\int_\theta P(D_{obs}|\theta)\pi(\theta) d\theta$ the marginal distribution of data. Then, the posterior distribution $\pi(\theta|D_{obs})$ evaluates to

$$\pi(\theta|D_{obs}) = \frac{P(D_{obs}|\theta)\pi(\theta)}{\int_\theta P(D_{obs}|\theta)\pi(\theta)d\theta}.$$

## Motivating example

For the purpose of illustrating our research problem over a motivating example, we first assume that the measurement apparatus does not distinguish the states which belong to a same terminal component (BSCC). Technically, the labels assigned to all states within one BSCC will be the same. Moreover, we assume that the labels can be read out only after the system runs for an extended period of time (long enough time to reach one of the BSCC's). Data from repeated experiments is then summarised into a histogram over BSCC's (labels). We are interested in inferring model parameters from such data.

**Non-identifiable parameters.** In the left example in Fig 1a, the parametric Markov chain has only one BSCC with three states, all labelled with label 'a'. Hence, all measurements detect label 'a', which contains no information about parameters. In the middle example in Fig 1b, each execution will either end up with label 'a', or with 'b', so it is possible to infer the

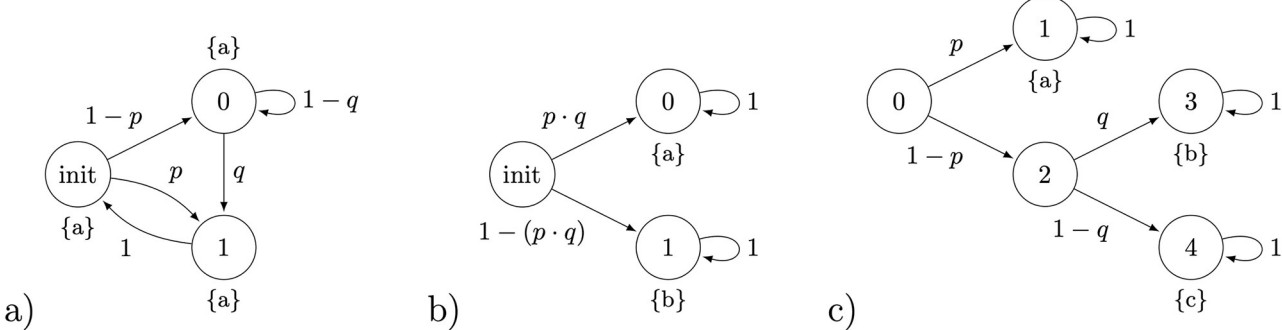

**Fig 1. Three examples of parametric Markov chains (pMC's) defined over the set of parameters $\mathcal{V} = \{p, q\}$.** a) The parameters cannot be identified because all states have the same label ($a$); b) The parameters cannot be uniquely identified, only their product $pq$ can be estimated from data. c) All parameters can be identified from repeated measurements of end states with output labels 'a', 'b', or 'c'.

probabilities of reaching the respective BSCCs. Yet, no matter how large the sample is, it is only the value of product $p \cdot q$ that can be inferred.

**Identifiable parameters.** In the third example in Fig 1c, only two parameters and three different labels (BSCCs) can be observed. Assuming that no measurement imprecision occurs (i.e. labels are correctly read-out from final states), a large enough sample size allows statistically inferring all parameters at a desired level of confidence.

At this point, limited sample sizes will require a careful propagation of uncertainty. To illustrate, consider an experiment where $N = 500$ measurements from model executions are observed to receive $N = 500$ samples of labels from the final state. We collect $N_a = 160$ samples which end up in a state labelled with $a$, and $N_b = 74$ in $b$ (and remaining $N_c = 266$ samples in $c$). Parameter inference may proceed with frequentist or Bayesian approach. In frequentist manner, parameter $p$ can be seen as a Bernoulli trial with success outcome $a$, and will be estimated to 0.32, with a margin at confidence level 95% equal to 0.04 (using the standard normal approximation of the binomial outcomes, more in Section Methods) hence $p \in C_0 = [0.28, 0.36] = I_p$. Regressing parameter $q$ will give an estimate $N_b/(N - N_a)$, as it can be seen as $N_b$ successes out of all outcomes that did not end up in $a$. However, determining the accompanying confidence interval for $q$ will depend on the number of samples but should also account for the randomness of outcome $N_a$.

One way to tackle this is to estimate the confidence intervals for a 'meta-parameter' $(1 - p) \cdot q$ and subsequently infer the margins for $q$. From two constraints: $p \in C_0$ and $(1 - p) \cdot q \in C_1 = [0.117, 0.179]$, we may deduce that whenever $p \in I_p$, and $q \geq \min(C_1)/\max(1 - I_p)$, and $q \leq \max(C1)/\min(1 - I_p)$, the parameter values will be consistent with the constraints read from data, that is, $p \in C_0$ and $(1 - p)q \in C_1$. The resulting rectangle $I_p \times I_q$ is depicted with dashed lines in Fig 2. Such result enjoys the standard interpretation of uncertainty in the frequentist sense: at 90% confidence level, the resulting confidence interval (CI) $I_p \times I_q$ contains the true parameter point $(p, q)$. Notice that, while the CI for each of the meta-parameters is derived for 95% level, this only means that the chance of not containing the true parameter value in *one of* the two created CI is at most 5%. The chances of missing to contain the true parameter value in *none of* the two created CI will increase, yet remain bounded: by Boole's inequality, $P(p \notin I_p \vee q \notin I_q) \leq P(p \notin I_p) + P(q \notin I_q)) \leq 10\%$. Generally, the CI for multiple parameters will require a correction for simultaneous confidence intervals; we will use a conservative extension of Bonferroni correction for testing multiple hypotheses [33]. In our example, to achieve an overall confidence of 95%, we would derive both parameters at 97.5% each.

A less conservative estimate of confidence region is possible through employing the third constraint $(1 - p)(1 - q) \in C_2$, from the proportion of data ending up in state '4'. A characterisation of the viable set of parameters (confidence set) respecting all three algebraic constraints is shown as the green area in Fig 2 (left). The obtained result will not depend on prior knowledge of parameters, and the Central Limit Theorem ensures that, given a sufficiently large sample size, the sample mean will provide a reliable estimate of the population mean, such that the desired level of statistical accuracy can be achieved for the estimate. It is worth noticing that one may analogously obtain the credible sets by inferring the introduced meta-parameters with a Bayesian approach (by using multinomial-Dirichlet conjugation).

For any given Markov chain, the described back-propagation of CIs involves the computation of rational functions for reaching respective BSCCs (e.g. $(1 - p)q$ for reaching 'b'), which easily becomes non-trivial with increasing the model size. In this work, we propose how to obtain these rational functions for arbitrarily complex Markov chains by leveraging the existing verification tools. Then, we show how they can be used to improve parameter inference procedures.

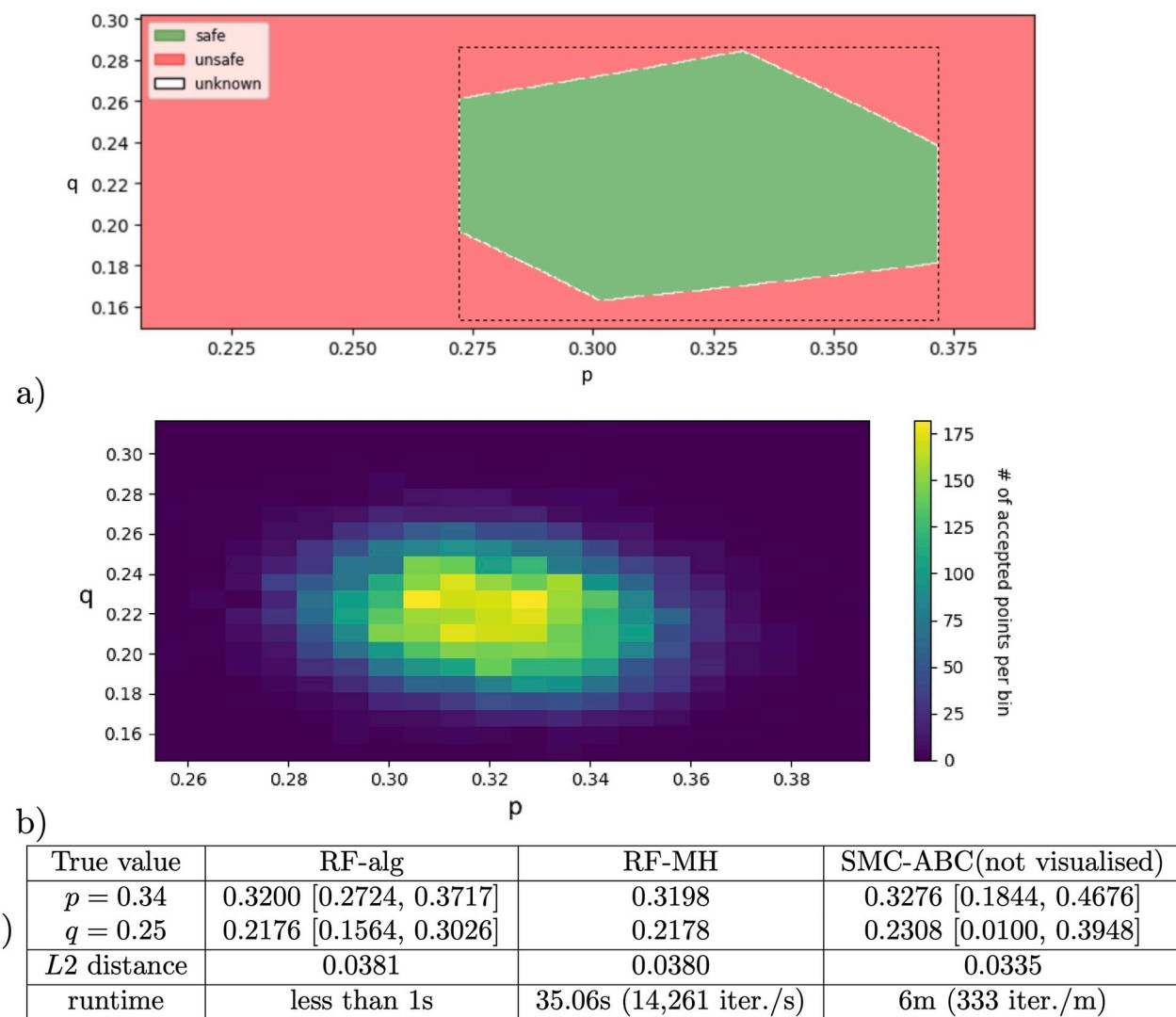

**Fig 2.** Parameter synthesis and inference for the motivating example in Fig 1(right) with $N = 500$ samples: (a) Green area represents parameter values agreeing with data at overall 95% level of confidence, obtained with parameter synthesis (RF-ref) for constraints $p \in C_0$, $(1 - p)q \in C_1$, $(1 - p)(1 - q) \in C_2$ (each constraint derived at 98.33% confidence level), with a space refinement algorithm (result shown in figure covering 99.99% of the domain). Dashed lines: intervals obtained by naive propagation of minimum and maximum values through algebraic manipulations of the first two constraints. (b) Results of Bayesian sampling-based inference performed with the exact likelihood function available (RF-MH). Values range from 0 (blue) to 175 (yellow) accepted points per bin. (c) Summary of parameter inference results for the two methods RF-alg, RF-MH, and a likelihood-free implementation SMC-ABC (not visualized). The $L2$ distance with respect to the true parameter values is given together with the respective runtimes.

- First, solving the obtained system of algebraic inequality constraints that are non-linear generally amounts to characterising a possibly non-convex space of parameters. To this end, we propose and implement a parameter synthesis procedure based on automated counterexample-guided space refinement (RF-ref). For the simple example in Fig 1 (right), a characterisation of the viable set of parameters (confidence set) respecting all three algebraic constraints through an approximation with hyper-rectangles is shown as the green area in Fig 2 (left).

- Second, in case of identifiable parameters, rational (likelihood) functions can be used to efficiently obtain a single estimate point by optimisation.

- Finally, in general, it is a challenge to achieve the scalability of parameter synthesis with counter-example guided refinement, as the number of dimensions increases. Different to parameter synthesis approaches based on the approximation of confidence (credible) sets with hyper-rectangles, Bayesian inference schemes sequentially sample the chain parameters and approximate their posterior distributions. In Fig 2 (right), we visualise the result of one such scheme for the same example. It provides an empirical quantification of uncertainty with respect to its closeness to data, however, the results generally depend on a number of hyper-parameters used in the algorithm (e.g. the length of the simulation, choice of priors, choice of perturbation kernels) that cannot be easily optimised or interpreted. As sampling-based Bayesian schemes involve computing the likelihood of data observations for each sampled value, two variants will be considered. First, when the likelihood is pre-computed as a rational function over the chain's parameters (e.g. $(1 - p) \cdot q$ for reaching label 'b' in our example), and second, the case when likelihood has to be approximated. The approach with the exact likelihood is potentially more accurate since it uses the true likelihood function and reduces uncertainty. Moreover, it is potentially more efficient, because evaluating rational functions is generally faster than statistically sampling the chain many times for each sampled parameter value.

In summary, we propose and implement how to compute the *exact* likelihood for given data in terms of rational functions over parameters of the MC, and how these rational functions can be used to: (i) efficiently compute the parameter points through maximising data likelihood (RF-opt method), (ii) compute the viable space of parameters complying with the data in the sense of traditional interpretations of uncertainty at a desired confidence level (RF-alg and RF-ref method), (iii) use rational functions to reduce uncertainty and boost scalability, through invoking them within an MCMC parameter inference scheme. The performance of these methods is compared with the likelihood-free Bayesian inference algorithm combining sequential Monte Carlo and approximate Bayesian computation (SMC-ABC). The table in Fig 2 illustrates the different results obtained for the motivating example, confirming that the approaches using the pre-computed rational functions provide more accuracy, precision and efficiency. In addition, the refinement-based approaches (RF-ref and RF-alg) guarantee an exact interpretation in terms of confidence intervals.

## Methods

In Fig 3, we show a workflow implementing the proposed methods for parameter search for a pMC with multiple BSCCs, where steady-state data observations are available. The methods differ in terms of whether the rational functions characterising the satisfaction probability of each among the multiple properties for reaching each of the BSCCs are available. In the implementations, we leverage existing tools PRISM [16] and Storm [17] to obtain the rational functions. All methods presented in this paper that use rational functions (RF-opt, RF-ref, and RF-MH) are implemented in the tool DiPS [34] (https://github.com/xhajnal/DiPS). The rational functions can be used for three different methods:

(i) *confidence level* (RF-alg, RF-ref): The (experimental) data are used as thresholds for constraining the rational functions for desired confidence intervals, resulting in a set of algebraic constraints. The resulting algebraic constraints are then employed to explore the parameter space for which the chain behaviour agrees with the observations. The algebraic constraints are finally resolved either with region generation and refinement with the help of theorem provers (RF-ref) (recall Fig 2 (left) from the motivating example).

(ii) *optimisation* (RF-opt): The values of parameters are found, such that the rational functions are closest to the data observation (in terms of least squares distance (L2)).

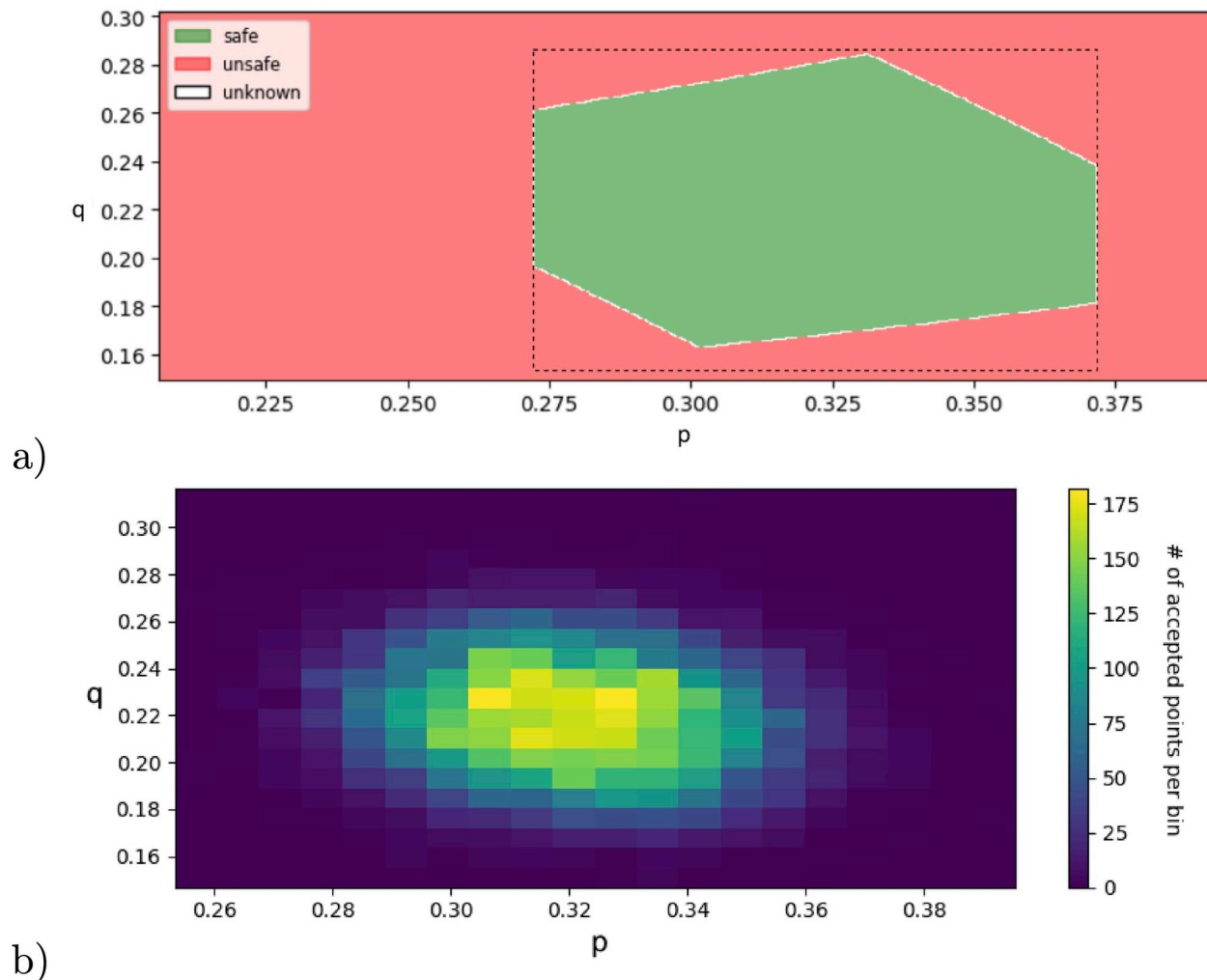

a)

b)

**Fig 3. Three classes of methods for parameter inference for DTMCs with steady-state data measurements (explained in Section Methods).**

(iii) *Sampling-based inference with exact likelihood* (RF-MH): Model parameters are sampled in a Metropolis-Hastings scheme, and the rational functions are used to evaluate the exact likelihood for each sampled parametrisation.

Finally, for the purpose of comparison, we also implement an approach where the rational functions are not employed, and the likelihoods are instead approximated by statistical sampling:

(iv) *Likelihood-free MCMC* (SMC-ABC): The parameters are sampled with sequential Monte Carlo (SMC) scheme, and the likelihood is approximated with the Approximate Bayesian Computation (ABC) algorithm.

### Rational functions as symbolic expressions for measured properties

In the motivating example, the distribution among the BSCCs as polynomial expressions over model parameters is captured by the polynomials, $p$, $(1 - p)q$ and $(1 - p)(1 - q)$, respectively.

In general,

$$\text{let } F_k(\mathcal{V}) \in \mathsf{Rat}(\mathcal{V}) \text{ be such that}$$
$$\text{for all } \theta \in \mathbb{R}^{|\mathcal{V}|}, \quad F_k(\theta) = \mathsf{P}(\mathcal{M}(\theta) \models FG(B_k)),$$

that is, $F_k(\mathcal{V})$ is a rational function over variables $\mathcal{V}$, exactly characterising the reachability (expressed as the PCTL property *Finally Globally*) of a BSCC uniquely labelled with $B_k$ in a Markov chain $M_\mathcal{V}$. We omit subscript $_\mathcal{V}$ when clear from the context. Notice that the formula does not involve the information obtained from data—it refers to the probability of eventually reaching a BSCC $B_k$, as a function of parameters of the chain.

## Data

We assume $N$ experiments of sufficient length in which we can observe which BSCC has been reached. Denote by $X_i \in [0, n]$ the outcome in experiment $i = 1, \ldots, N$ describing which BSCC has finally been reached by the system. The uncertainty can be quantified through margins in different ways: (i) *confidence intervals*, thus providing an interpretation in a frequentist manner, and (ii) *credible sets* utilising the multinomial-Dirichlet conjugate priors. In the first approach, we estimate the probability of reaching each of the BSCCs in a standard way: at confidence level $(1 - \alpha_k)$ for all $B_k \in BSCC(\mathcal{M})$, $\mathsf{P}_{M(\mathcal{V})}(FG(B_k)) \in \hat{x}_k \pm m_k$, where $\hat{x}_k := \frac{X_k}{N}$ is the point estimate for the probability of eventually reaching BSCC $B_k$, and $m_k$ the corresponding margin of the confidence interval at level $(1 - \alpha_k)$. As pointed out in the motivating example, in order to claim an overall confidence level $1 - \alpha$, the CI for multiple parameters will require a correction for simultaneous confidence intervals; we will use a conservative extension of Bonferroni correction for testing multiple hypotheses [33], and hence choose $\alpha_k := \alpha/(n + 1)$, for $k = 0; 1, \ldots, n$, which we explain in Lemma 1 below. Improvements are possible with different corrections tailored to the multinomial proportions [35]. As we have Binomial proportion data with large sample size, we use the Agresti-Coull method for confidence intervals [36] in the experiments instead of the standard Wald method, that frequently fails to achieve the nominal coverage level [37]. For other conditions, e.g. Wilson [36], Jeffreys [36], Clopper-Pearson [36], or Rule of three [38] method can be used. Bayesian estimation of credible sets is possible through updating the prior $\mathsf{Dir}(\alpha_1, \ldots, \alpha_n)$ to the posterior is $\mathsf{Dir}(\alpha_1 + x_1, \ldots, \alpha_n + x_n)$, where $x_i = X_i/N$, however, the obtained credible set will not be unique, as the choice of prior will affect the posterior.

**Lemma 1** (*Correction for inferring multiple CI's*) Let $\{\theta_i\}_{i=1}^n$ be the true parameters of a multinomial distribution, $\{\alpha_i\}_{i=1}^n$ be such that $\alpha_i \in (0, 1)$ and $\Sigma_i \alpha_i = \alpha \in (0, 1)$, $\{I_i\}_{i=1}^n$ a family of intervals on $[0, 1]$. Then, if each interval satisfies $P(\theta_i \in I_i) \geq 1 - \alpha_i$, it also holds that $P(\bigcap_i \theta_i \in I_i) \geq 1 - \alpha$.

The proof follows by bounding the probability of the complementary event via Boole's inequality. For simultaneous confidence intervals for multinomials, the presented Bonferroni correction is often conservative, especially as the number of bins (classes) grows, due to correlated outcomes. Improvements are possible with different corrections tailored to the multinomial proportions [35].

## Methods using rational functions

**Parameter synthesis with space refinement (RF-ref).** Inferring the parameters at a desired confidence level can be obtained by solving the conjunction of algebraic inequalities

for parameters of the chain:

$$\bigwedge_{k=0}^{n} (F_k(\mathcal{V}) \in [\hat{x}_l, \hat{x}_u](k \mid \texttt{data})), \tag{1}$$

expressing that each of the BSCCs is reached with a probability within a confidence interval obtained from data. Every parameter evaluation $\theta \in [0,1]^{|\mathcal{V}|}$ such that the constraints hold, belongs to our goal *viable set* $\Theta$, and vice versa. A single point estimate may be satisfactory in some cases, and the method of *optimisation* refers to finding the point in parameter space closest to the data observations (corresponding to the maximum likelihood estimate). However, to account for the uncertainties in the inference process, we wish to characterise the points complying with the derived confidence intervals as closely as possible (i.e. the green region in Fig 2 left). Sampling-based techniques allow exploring the parameter space for a finite number of points, hence providing no global guarantees. On the other hand, in our implementation, we perform a global search of the parameter space: we pass a query $\exists \theta \in (\mathcal{V} \mapsto [0,1]^{|\mathcal{V}|})$, such that $\bigwedge_{k=0}^{n} F(k)(\mathcal{V}) \in [\hat{x}_l, \hat{x}_u](k \mid \texttt{data})$ to an SMT solver, such as Z3 [39] or dReal [40], or to an interval arithmetics solver such as Python `mpmath` library. Then, depending on the outcome, we further refine the parameter space in CEGAR-like (counterexample-guided abstraction-refinement) fashion into

- $\Theta_{green}$, *safe* or "green" regions, where the constraints are met,

- $\Theta_{red}$, *unsafe* or "red" regions, where the constraints are not met,

- $\Theta_{white}$, *unknown* or "white" regions, where the constraints may hold or not,

the idea of which is also used by existing tools, such as Prophesy [30]. For each parameter evaluation in a *safe* region, the formula holds because the negation of the constraints is not satisfiable. For each parameter evaluation in the *unsafe* region, the constraints are not satisfiable. The unknown region is not refined yet or it contains both, parameters for which the formula holds and for which it does not hold. In our implementation, we use a naive splitting into two halves along the dimension with the largest range. This split occurs when the given region is proven to be neither safe nor unsafe. As the main stopping criterion, we introduce the parameter coverage, such that the fraction of the explored parameter space and the whole parameter space is larger than coverage: $\Theta_{green} + \Theta_{red} > $ coverage.

For the evaluation, we are using the sampling-guided version, which samples rectangles before refinement to avoid expensive solver calls, z3 as the solver, parallel version with six cores—able to solve up to six rectangles simultaneously, and alg2 (DiPS setting), which encodes simple splitting without passing examples/counterexamples of satisfaction.

**Sampling-based inference with exact likelihood (RF-MH).** In our problem setup, the analytical form of the posterior distribution for parameters of the chain (e.g. $p$ and $q$ in the motivating example) is generally not available and hence different additional assumptions and/or approximations must be used to approximate the posterior with Bayesian inference.

We implement a basic Metropolis-Hastings scheme [41], a Markov chain Monte Carlo algorithm, where we employ the knowledge of precomputed rational functions to evaluate the likelihood in each newly sampled parameter point. Starting in a selected initial point $\theta_{init}$, Metropolis-Hastings walks in the parameter space for a selected number of iterations. In each iteration, a *transition function* picks a new point $\theta'$ in the parameter space by perturbing the current point $\theta$ with an adjustable variation value. Next, likelihoods of these two points, $\theta$ and $\theta'$, are compared (we consider non-informative uniform distribution and the evidence strikes out—see Definition 4), and if the likelihood of the new point is larger $P(D_{obs}|\theta') > P(D_{obs}|\theta)$,

we *accept* the proposed point and move in the parameter space. If the likelihood is smaller, there is a small probability of accepting the new point, $\theta'$, anyway—this helps to avoid local optima. Lastly, if the proposed point is *rejected*, we select the current point, $\theta$, for the next iteration. The set of accepted points is used to approximate the posterior distribution. In the one- or two-dimensional case, the space is rectangularised into a selected number of `bins`, and each bin is visualised with a colour grade based on the number of the accepted point s within the bin—see Fig 2 (right). For more dimensions, a scatter-line plot showing each of the accepted points is created—see Figs 4c and 5c.

## Likelihood-free sampling-based inference

We combine the Sequential Monte Carlo sampling and the Approximate Bayesian Computation (ABC) algorithm to implement a likelihood-free inference scheme to be compared to other proposed methods (SMC-ABC). SMC, firstly proposed by Del [42], addresses the issues of Metropolis-Hastings, by being easily parallelisable and less likely to fall into a local maximum or minimum. The ground idea is to, instead of having one particle moving in its parameter space, use a number of particles moving independently. In each iteration, it then mutates parameter candidates through a series of *perturbation kernels* and selects parameter candidates for the next iteration, taking into account their weights.

The Approximate Bayesian Computation (ABC) method [43] is a widely used likelihood-free method for approximating posterior distribution, useful in scenarios in which the likelihood has no analytical form, or the analytical form is expensive to be evaluated. In the context of the problem considered in this paper, it applies when rational functions are not obtainable due to the large size of the Markov chain or when they are too expensive to evaluate numerically. Instead of estimating the likelihood $P(D|\theta)$ directly, we define a distance measure $\delta(D_1, D_2)$ where $D_1$ and $D_2$ denote observable data. Given a parameter candidate $\hat{\theta}$ that specifies a model $\mathcal{M}_\theta$, the ABC algorithm accepts $\hat{\theta}$ if a simulation run on $\mathcal{M}_\theta$ delivers observable data $D_{obs}$ such that $\delta(D_{obs}, D_{sim}) < \epsilon$, where $\epsilon \in \mathbb{R}_{\leq 0}$ is the distance threshold. ABC algorithm can be used together with Markov chain Monte Carlo algorithms (ABC-MCMC [44, 45]), or with Sequential Monte Carlo sampling algorithms (ABC-SMC [46, 47]). We implement the latter. We select uniform distribution as the prior because it is less likely to propagate false beliefs to the subsequent Bayesian inference [26]. The perturbation kernel is the *component-wise uniform kernel* [48]. At each perturbation, we use a uniform distribution with boundaries adjusted by the previously sampled parameter values. For multi-dimensional parameters, each parameter component has an independently adjusted uniform kernel. In this paper, we use a population of 500 parameter values to estimate the posterior. The SMC algorithm then mutates the population through 5 perturbation kernels. We visualise the results by showing the found parameter point and a 95% highest posterior density (HPD) interval around it for each dimension (Figs 4c and 5c).

## Case studies

In this section we present four case studies: nested branching, honeybee collective stinging response, SIR model, and Zeroconf protocol. We have obtained synthetic data showing the reachability distribution among the BSCCs by simulating the Markov Chain using a selected true parametrisation. Finally, we compare the results for different methods: using rational functions (RF-opt, RF-MH, RF-alg) and without using them (SMC-ABC). The evaluation was run on a tower PC Skadi—64bit Ubuntu 20.04.2, i9-9900K CPU, 32GB RAM, SSD disk. Run-time comparison of the methods is provided in Table 1. The software used for the analysis and plotting is publicly available on GitHub at the repository https://github.com/xhajnal/DiPS. All

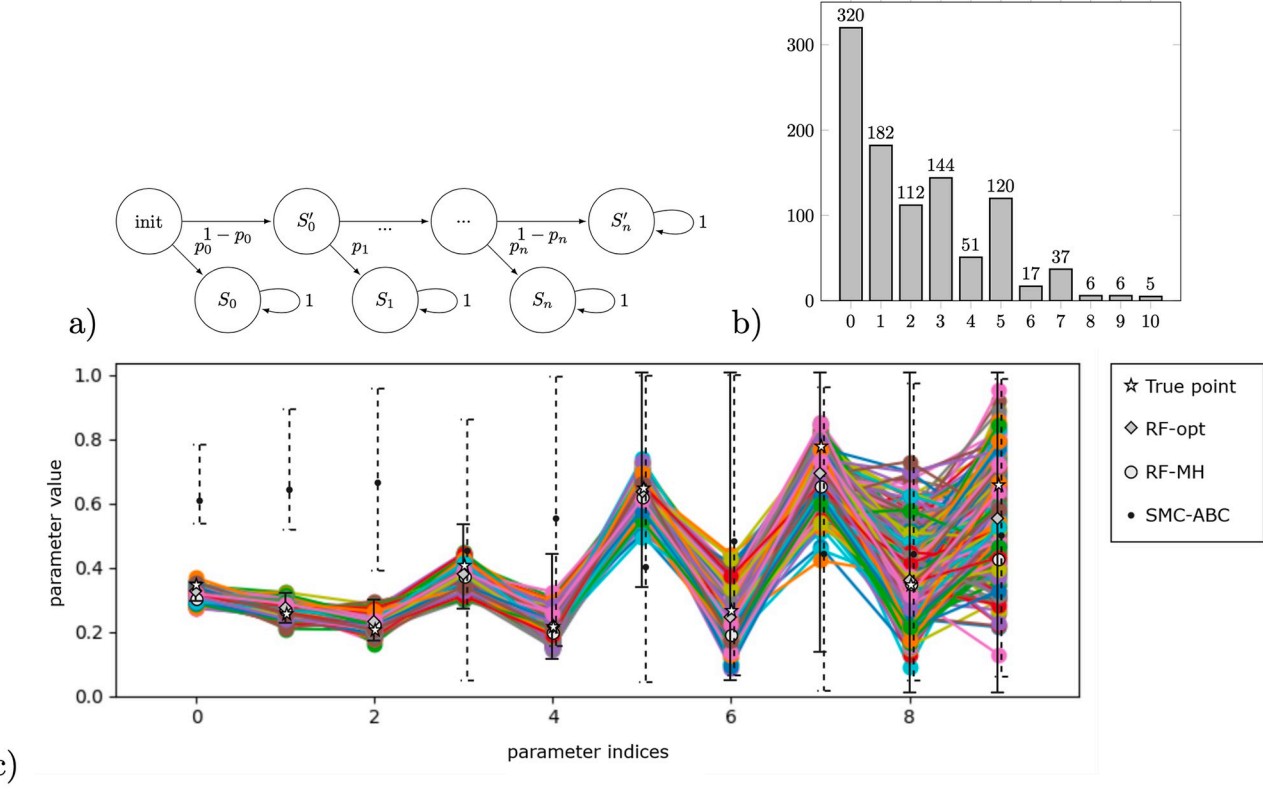

**Fig 4.** a) Nested branching pMC for 10 parameters and 11 BSCC's ($n = 9$). b) Data histogram of reaching the respective BSCCs as a result of 1000 simulations with true parameter values visualised below. c) Visualisation of inference results obtained by different methods, x-axis representing the index of parameter, (i) 95% confidence intervals obtained by the interval propagation (RF-alg, black error bars), (ii) accepted parameter points obtained by sampling-based inference using exact likelihood (RF-MH, one colour displays one accepted point). We visualise a set of accepted points as a result of 12,114,821 iterations, sample size = 1000, trimming first 25% of 477 accepted points, obtained in 1h processor time using Skadi, (iii) HPD estimate at 95% confidence level (dashed error bars), obtained by SMC-ABC in 1h45min using Skadi. 95% HPD credible sets, obtained by likelihood-free sampling based inference (SMC-ABC, dashed error bars) in 1h45min processor time using Skadi.

the input and output files are publicly available at the Zenodo repository https://zenodo.org/record/7900258#.ZFzbIexBwqs.

## Nested branching

**Model description.** We expand the motivating example with identifiable parameters, shown in Fig 1c, to a general pMC with $n$ parameters and $n + 2$ BSCCs, shown in Fig 4a. The model describes a branching process in which there are 2 possibilities at each step: either a BSCC is reached, or the next branching is reached. Here, we consider a branching process with $n + 1 = 9$ parameters and $n + 2 = 11$ BSCCs. The data for the experiments are generated by simulating the pMC with previously chosen true parameters, shown in Fig 4b.

**Results.** Notice that the number of parameters in the model is equivalent to the number of BSCCs. Moreover, due to the specific structure of the chain, all parameters are identifiable from the steady-state data. Still, there is a challenging propagation of uncertainty to handle, because reaching some BSCCs requires making multiple transitions in the chain, each of which involves at least one parameter. In Fig 4c, we visualise the true parameter values used to synthesise data, the best parameter estimates achieved with different algorithms,

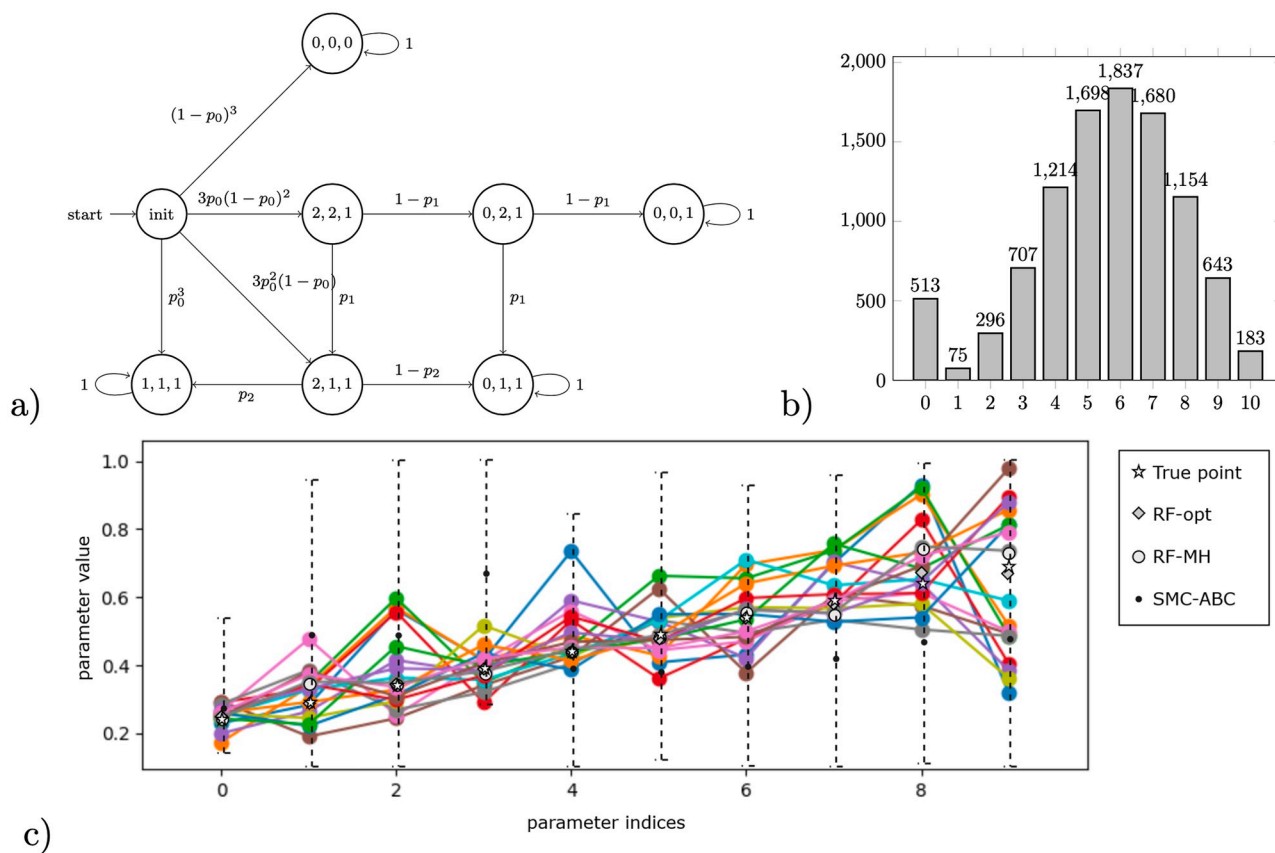

**Fig 5.** a) Honeybees pMC for population $n = 3$. Notice that the distribution among possible final states (BSCC's) is a list of $2n$-degree multivariate polynomials over model parameters, e.g. the probability of reaching state $(0, 1, 1)$ is $3p_0^2(1 - p_0)(1 - p_2) + 3p_0(1 - p_0)^2(p_1(1 - p_2) + (1 - p_1)p_1)$. b) Data histogram of reaching the respective BSCCs (number of stinging bees) obtained from 10,000 simulations using true parameter values visualised in c). c) Visualisation of inference results obtained by different methods, x-axis representing the index of a parameter ($p_0, p_1, \ldots, p_9$): (i) accepted parameter points obtained by sampling-based inference using exact likelihood (RF-MH, one colour displays one accepted point). We visualise a set of accepted points as a result of 216,616 iterations, sample size = 1000, trimming first 25% of 25 accepted points, obtained in 3 hours processor time using PC Skadi, (ii) 95% HPD credible sets, obtained by likelihood-free sampling based inference (SMC-ABC, dashed error bars) in 3h45min processor time using Skadi.

as well as the respective confidence intervals. Runtimes are reported in Table 1. RF-ref is not shown in Fig 4c as it reached the coverage of 0.99946 within 158s and discovered no safe rectangles (obtained using Skadi, sampling-guided refinement with z3 solver, 6 parallel cores).

**Table 1. Runtimes of used methods with Skadi given in interations per second.** RF-ref time to reach standard coverage 0.95 using alg2, z3 solver, with 6 parallel cores. RF-alg runtimes are not shown as they are not implemented for the general case. Solving a system of nonlinear inequality constraints is beyond the scope of this manuscript. Timeout (TO) 1 hour.

| | RF-opt | RF-ref | RF-MH | SMC-ABC |
|---|---|---|---|---|
| nested branching $n = 2$ | 0.0101s | 0.28s | 14,261.3 iter/s | 333 iter/min |
| nested branching $n = 10$ | 0.0228s | 0.76s | 21.7 iter/s | 49.8 iter/min |
| honeybees $n = 10$ | 3.4023s | TO | 20.1 iter/s | 1.25 iter/min |
| zeroconf $n = 4$ | 0.0096s | 1.85s | 13,320 iter/s | 660.06 iter/min |
| SIR $n = 5$ | 0.0144s | 18.59s | 3,572.7 iter/s | 247.52 iter/min |

**Discussion.**    All methods using the pre-computed rational functions provide better accuracy with respect to the SMC-ABC implementation. Moreover, the RF-MH method improves the precision for parameters $p_4, \ldots, p_9$. Notice that the uncertainty increases for increasing the index of parameters, that is, the margins for the parameters with larger indices become larger. In particular, the parameter $p_0$ is inferred with lowest uncertainty, and parameter $p_9$ with largest uncertainty. This is because less samples end up in the 'later' BSCCs (Fig 4b), and, consequently, the sample size for inferring 'later' parameters is smaller. For instance, while only the data for ending up in the last two BSCCs $S_9$ and $S'_9$ tells us something about parameter $p_9$, *all* other outcomes inform us about parameter $p_0$. In Fig 4c, we see that the estimates obtained with SMC-ABC are far from the true points and have large credible sets. This is because SMC-ABC explores all parameters at once and treats them equally, rather than taking into account that uncertainty is higher for later parameters. Enriching the SMC-ABC method with a preliminary uncertainty analysis would allow us to explore the parameters in a more efficient way; such analysis is beyond the scope of this paper which we leave to future work.

**Reproducibility.**    The sampled data and the PRISM model are available in Zenodo together with the text file including the analysed PCTL properties (see `branching_model_10_data_1000_samples.txt`, `branching_model_10.pm`, `branching_model_10.pctl`).

## Social feedback in honeybee colonies

We present a case study modelling a real-world phenomenon of social feedback mechanism in honeybee colonies. The presented model, which is similar to the nested branching model, was first introduced in [28], and its adapted variant was validated with respect to experimental data in [29].

In the field of biology, experts often excel at describing the qualitative aspects of a phenomenon well and speculate about their underlying mechanisms. However, precise statements are hindered by the lack of corresponding quantitative explanations. This compelling case study demonstrates how our method enabled researchers to not only model the qualitative aspects of decision-making as a Markov Chain, but also quantitatively the underlying mechanisms using data.

**Model description.**    After observing a threat in the environment, a bee in the colony may sting and consequently die. Each stinging bee releases an alarm pheromone, hence recruiting more and more bees to sting. However, if the aggressiveness keeps increasing with the amount of pheromone present, the colony may be extinct. The mechanism as to how precisely the trade-off between efficient defence and maintaining workers' force is established is not known to date [49].

Consider a colony of $n$ bees and an experiment ending with a number of stinging bees ranging from 0 to $n$. Following [28], the colony is modelled with a parametric discrete-time Markov Chain with $(n + 1)$ BSCCs encoding the population (number of stinging bees) at the end of the experiment. Each agent commits an action (stinging) with a certain probability, leading to its immediate death. Each individual bee is encoded by an integer representing its state, with the following encoding:

0: never stings,

1: stings and dies,

2: it does not sting without additional stimuli but may be recruited when the alarm pheromone is present.

Parameter $p_i$ alters the stinging probability based on the amount of alarm pheromone. For example, for a colony of $n = 3$ bees—see Fig 5a, the following pMC is constructed: a bee stings without any pheromone present with probability $p_0$, with one unit of pheromone available with probability $p_1$, and with two units of pheromone with probability $p_2$. Here we consider a semi-synchronous version of the model, where the first event of stinging (before sensing the alarm pheromone) is made synchronously (all of the bees decide at the same time), and all other stinging events are asynchronous (only one bee can sting in one time-step). We analyse a model of $n = 10$ bees and hence 11 BSCCs.

**Results.** Notice that, similarly to the model of nested branching, the number of parameters in the model is equivalent to the number of BSCCs. Moreover, due to the specific structure of the chain, all parameters are identifiable from the steady-state data. Yet, the model exhibits a challenging propagation of uncertainty, because reaching some BSCCs requires making multiple transitions in the chain, each of which involves at least one parameter.

Synthetic data obtained by 10000 simulations of the MC is shown in Fig 5b. As the underlying chain counts 69 states and rational functions are non-linear multivariate polynomials of order 21, the back-propagation of the algebraic constraints from confidence intervals for meta-parameters to the parameter of the chain (RF-alg) is not performed for this case study. Moreover, the counterexample-guided refinement (RF-ref) timed out at 1 hour. In Fig 5c, we visualise the results obtained with different methods together with the true parameter points.

**Discussion.** Our results confirm that the methods using the pre-computed rational functions (RF-opt and RF-MH) provide more accuracy with respect to the SMC-ABC implementation. Respective runtimes are reported in Table 1, indicating a significantly slower single iteration in the likelihood-free SMC-ABC implementation with respect to RF-MH. We explain this by the fact that SMC-ABC repeatedly simulates a chain of 69 states, each with at least 10 steps until reaching a BSCC in each iteration, while the RF-MH method evaluates the rational functions instead. Moreover, similarly to the nested branching example, we here again observe larger uncertainty for the last two parameters due to the small number of samples ending up in the respective BSCCs where $p_8$ and $p_9$ occur in the chain.

**Reproducibility.** The sampled data and the PRISM model are available in Zenodo together with the text file including the analysed PCTL properties (see `bees_10_data_10000_samples.txt`, `bees_10.pm`, `bees_10.pctl`).

## SIR

**Model description.** The *Susceptible Infected Recovered* (SIR) model is the most common stochastic model for predicting disease spread [50]. Each agent in a homogeneous, well-mixed population can be in one of three states: S, I, or R. The dynamics of the system are described by two reactions, where we denote the infection rate when meeting an infected individual by $\alpha$, and the recovery rate by $\beta$:

$$R_1 : \quad S + I \xrightarrow{\alpha} 2I$$
$$R_2 : I \xrightarrow{\beta} R$$

The stochastic dynamics is captured by a continuous-time Markov chain (CTMC), where each state enumerates the number of each of the agent types, as shown in Fig 6a. Since the stationary distribution of a CTMC is equivalent to that of its uniformised DTMC [51], we can apply our proposed workflow to the respective uniformised DTMC. While intuitively, uniquely identifying two parameters from more than two data points (BSCCs) can be possible, this case study will showcase a situation where only a linear subspace of parameters can be identified from steady-state data.

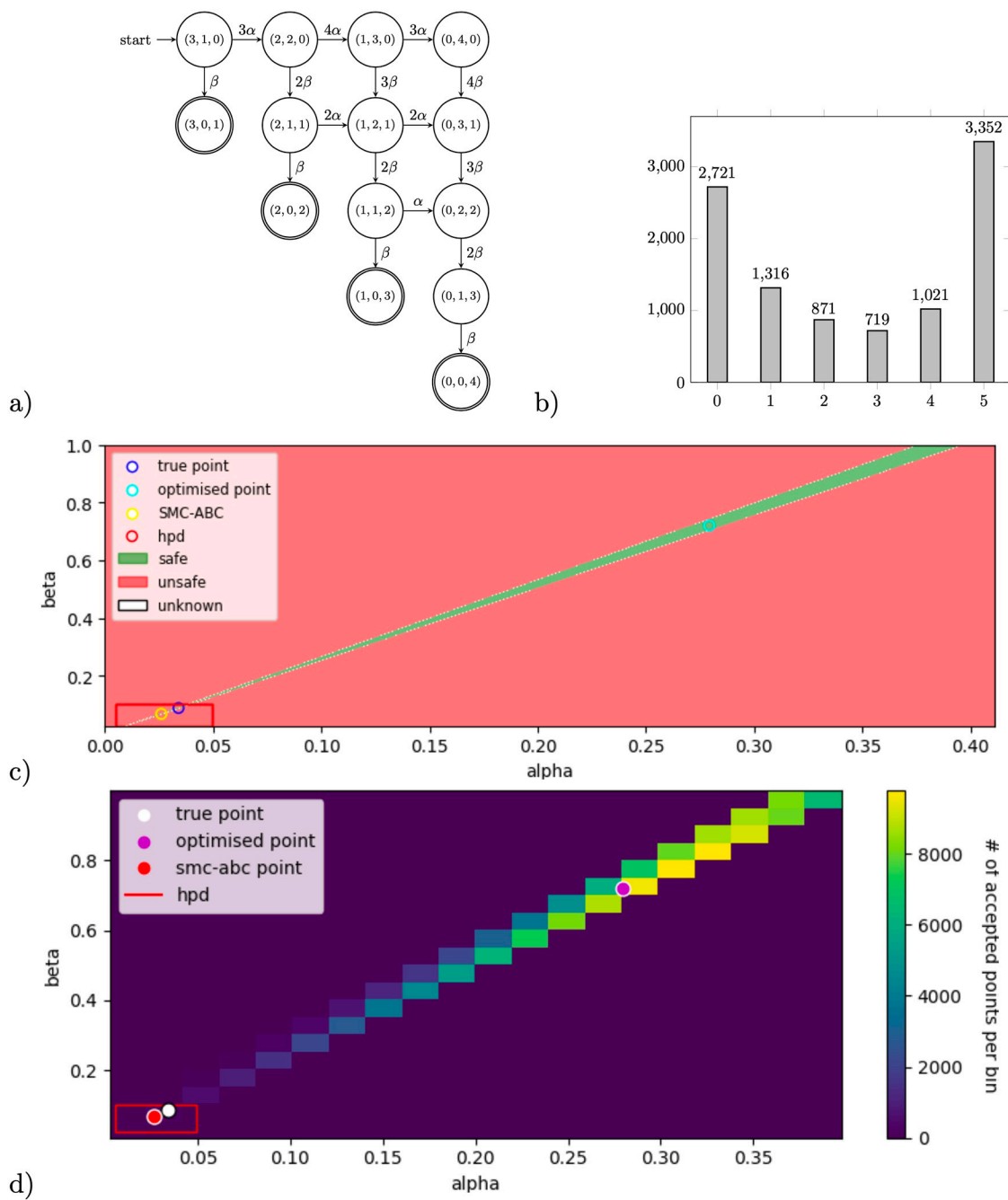

**Fig 6.** a) The pMC (continous-time) for a SIR model with $n = 4$ agents. b) Data histogram of reaching the respective BSCCs obtained by 10,000 simulations using true parameter values, in a SIR model with $n = 5$ agents. c) Visualisation of RF-ref results, where the green area shows parameters for which the rational functions fall within the intervals created from the data. The true point is shown as a blue dot, the result of RF-opt shown as a cyan dot, and the result of SMC-ABC shown as a yellow point. d) Visualisation of RF-MH results after 12,861,593 iterations, while trimming the first 25% of 228,240 accepted points obtained in 1 hour processor time using PC Skadi. The true point is shown as a white dot, the result of RF-opt shown as a purple dot, and the result of SMC-ABC shown as a red point. In both figures, c) and d), the red rectangle shows the 95% HPD credible set, obtained by SMC-ABC.

See Fig 6a for an example of a CTMC for $n = 4$ agents. In further analysis, we showcase a model of $n = 5$ agents. Since all rates in the CTMC for the SIR model are scalings of parameters $\alpha$ and $\beta$ (therefore, of form $k\alpha$ or $k\beta$ for some integer number $k \geq 0$), for simplicity, we choose a uniformisation rate to be the sum of the two possibly largest rates (which, for $n = 4$ agents amounts to $4\beta + 4\alpha$, and for $n = 5$ agents is $9\alpha + 6\beta$). Synthetic data is obtained by simulating the chain 10,000 times with parameters $\alpha, \beta = [0.034055, 0.087735]$, leading to the following probability distribution of eventually reaching respectively $1, \ldots, n$ infected agents: $[0.2721, 0.1316, 0.0871, 0.0719, 0.1021, 0.3352]$.

**Results.** In Fig 6c, we visualise the results of RF-ref as a green area representing all possible parameter combinations of $\alpha$ and $\beta$ for which the rational functions fall within the respective data intervals. Fig 6d shows the results of RF-MH as the number of accepted points for all parameter combinations. In both visualisations, the true parameter point and the results of RF-opt and SMC-ABC are shown as coloured dots.

**Discussion.** Intuitively, uniquely identifying two parameters from more than two data points (BSCCs) may be possible. Yet, this example showcases a situation where parameters $\alpha$ and $\beta$ are not uniquely identifiable from the given steady-state data observations. In this case, the traditional optimisation methods such as RF-opt, which provide only single-point estimates, can be far from the true point, and it becomes important to obtain global information of the space of parameters for which the model complies with the data (defined in Section Methods and following Eq 1, *viable* set of parameters $\Theta$). Note that in this example the single-point estimate of SMC-ABC is close to the true point. This result is obtained by taking an appropriate prior distribution, here *uniform*$(0, 0.1)$, which already provides information about the parameter ranges. The prior is chosen according to real-world applications of the SIR model, where parameters $\alpha$ and $\beta$ are reported to lie within this range [52].

In Fig 6c and 6d, we see that RF-ref and RF-MH methods both automatically capture the correlation between $\alpha$ and $\beta$ in the viable parameter space. The green region in Fig 6b visualises the viable space $\Theta$ as defined in Section Methods (following Eq 1), and the heat-map in Fig 6c shows its weighted version. Both the green area (method RF-ref), but also the heat-map (method RF-MH) suggest that data consistency is invariant with respect to *linear* scaling of parameters $\alpha$ and $\beta$. Moreover, we observe a widening of the green region for increasing values of $\alpha$ and $\beta$. Mathematically, whenever $\theta = (\alpha, \beta)$ instantiates a CTMC with the distribution among the BSCCs $\pi \in [0, 1]^{|BSCC(\mathcal{M})|}$, so will any scaling $\theta_c = (c\alpha, c\beta)$, for any $c \in \mathbb{R}_+$; this is a direct consequence of the fact that scaling all rates of a CTMC with the scalar $c$ preserves the transient distribution until the corresponding scaling of the time units. Furthermore, the viable parameter subspace $\Theta$ is closed upon linear scaling: whenever $\theta \in \Theta$ complies with the sample distribution $\hat{\pi}$ at a desired confidence level, so will any scaling $c\theta$, for any $c \in \mathbb{R}_+$. We support both these observations with the following lemma.

**Lemma 2** (*Invariant steady-state distribution upon linear scaling of model parameters*) Let $\mathcal{M}_\theta$ be a parametric CTMC such that its transition rates are linear combinations of a parameter vector $\theta \in \mathbb{R}^{|\mathcal{V}|}$, that is, they are of the form $q_j = \sum_i x_{i,j} \cdot \theta_i$, with $x_{i,j} \in \mathbb{R}$ denoting a linear coefficient of the $i$-th parameter $\theta_i$ in the $j$-th transition. Then, for any $c \in \mathbb{R}^+$, the CTMC $\mathcal{M}_{\theta \cdot c}$ obtained by scaling all parameters in $\theta$ with a factor $c$, is such that the transient probability distribution of $\mathcal{M}_\theta$ at time $t$ is exactly the same as of $\mathcal{M}_{\theta \cdot c}$ at time $tc^{-1}$, i.e. $P_\theta(t) = P_{\theta \cdot c}(tc^{-1})$. In particular, $\mathcal{M}_{\theta \cdot c}$ has the same steady-state distribution as the base model $\mathcal{M}_\theta$.

**Proof 1** *Scaling the parameters with $c$ gives the new transitions $\sum x_{i,j} \cdot c \cdot \theta_i = c \cdot \sum x_{i,j} \cdot \theta_i$. Therefore, the generator matrix in the new chain is $Q_{\theta \cdot c} = c \cdot Q_\theta$. Uniformisation of the base model with $r$ and the scaled model with $r_c = c \cdot r$ results in the transition matrices $P_\theta$ and $P_{\theta \cdot c}$, such that $P_{\theta \cdot c} = I + \frac{1}{r_c} Q_{\theta \cdot c} = I + \frac{1}{r \cdot c} c \cdot Q_\theta = I + \frac{1}{r} Q_\theta = P_\theta$. Accordingly, the transient probability*

*distributions are defined by $P_\theta(t) = \sum_{n=0}^{\infty} P_\theta^n e^{-rt} \frac{(rt)^n}{n!}$ and $P_{\theta \cdot c}(t) = \sum_{n=0}^{\infty} P_\theta^n e^{-c \cdot rt} \frac{(c \cdot rt)^n}{n!}$. It directly follows that $P_{\theta \cdot c}(tc^{-1}) = \sum_{n=0}^{\infty} P_\theta^n e^{-c \cdot rtc^{-1}} \frac{(c \cdot rtc^{-1})^n}{n!} = \sum_{n=0}^{\infty} P_\theta^n e^{-rt} \frac{(rt)^n}{n!} = P_\theta(t)$, which shows the equivalence of transient distributions of $\mathcal{M}_\theta$ and $\mathcal{M}_{\theta \cdot c}$ at time points $t$ and $tc^{-1}$ respectively. It further follows that in the long run, so for $t \rightarrow \infty$, and thus also $tc^{-1} \rightarrow \infty$, $P_\theta(t) = P_{\theta \cdot c}(t)$. Therefore, $\mathcal{M}_\theta$ and $\mathcal{M}_{\theta \cdot c}$ have the same steady-state distribution.*

Since the rates in the CTMC model of SIR spread are indeed linear combinations of parameters $\alpha$ and $\beta$, it follows that, for any $c \in \mathbb{R}_+$, parametrisations $\theta = (\alpha, \beta)$ and $\theta_c = (c\alpha, c\beta)$ will induce two different CTMCs with exactly the same steady-state distribution s, hence explaining the linear viable subspace. Another direct corollary of the above lemma is that the viable parameter subspace $\Theta$ is closed upon linear scaling. It thus holds that, whenever a parameter region $\Theta' \subset \Theta$ complies with sample distribution $\hat{\pi}$ at a desired confidence level, so will any scaling $c\Theta'$, for any $c \in \mathbb{R}_+$, hence explaining the widening of the green area along the axes denoting $\alpha$ and $\beta$.

**Reproducibility.**   The sampled data and the PRISM model are available in Zenodo together with the text file including the analysed PCTL properties (see `SIR_5_data.txt`, `sir_5_1_0.pm`, `sir_5_1_0.pctl`).

## Zeroconf

**Model description.**   We use Zeroconf, a model of a well-known computer network protocol, to demonstrate another scenario where parameters are not uniquely identifiable from the steady-state measurements; more concretely, we show that our methodology allows to automatically capture non-linear dependencies characterising the viable parameter space.

Zeroconf [53] is a computer network protocol built to provide a new device within the network with an IP in a lossy environment without intervention from other network operators. The device randomly selects an IP and sends $n$ probes containing the message of the selected IP to all network nodes to find out whether the selected IP is in use, which resets the protocol with a different IP, or the IP is vacant and select it as its own. Parameter $p$ describes the probability of a message to be lost (no reply and time out) while parameter $q$ is the probability of the selected IP being in use—this models network occupancy. In the model, state *OK* describes a situation where a unique IP is selected, while state *Failed* indicates the selection of a non-unique IP, which can only happen if all probes are lost.

We obtained synthetic data for a model with $n = 4$ probes, shown in Fig 7a, with the chosen parameter point $[p, q] = [0.105547, 0.449658]$. After 10,000 simulations, we got the following probabilities to reach the two states $[OK, Failed] = [0.9999, 0.0001]$, see Fig 7b. While the generated data shows that for relatively high probability $p$ a message will be lost, using 4 probes induces the correct behaviour with high probability (99.99%). Our methodology allows one to characterise the global set of parameters for which the correct behaviour is obtained with probability 99.99%. In practice, such analysis may inform the protocol design choice as to how many probes to use when $p$ and $q$ change, or are given in ranges of possible values.

**Results.**   In Fig 7c we visualise the results of RF-MH method, and in Fig 7d we visualise the results of RF-ref, as a green area approximating the viable parameter space. In both visualisations, the true parameter point and the results of RF-opt and SMC-ABC are shown. In addition, we show the 95% HPD credible sets obtained from SMC-ABC method. The sampled data and the respective PRISM model are available in Zenodo together with the text file including the analysed PCTL properties (see `zeroconf_4_data.txt`, `Zeroconf_4.pm`, `Zeroconf_4.pctl`).

**Discussion.**   In this example, parameters $p$ and $q$ are not uniquely identifiable. This is expected, because only one data measurement is given (reaching one of the BSCCs, because

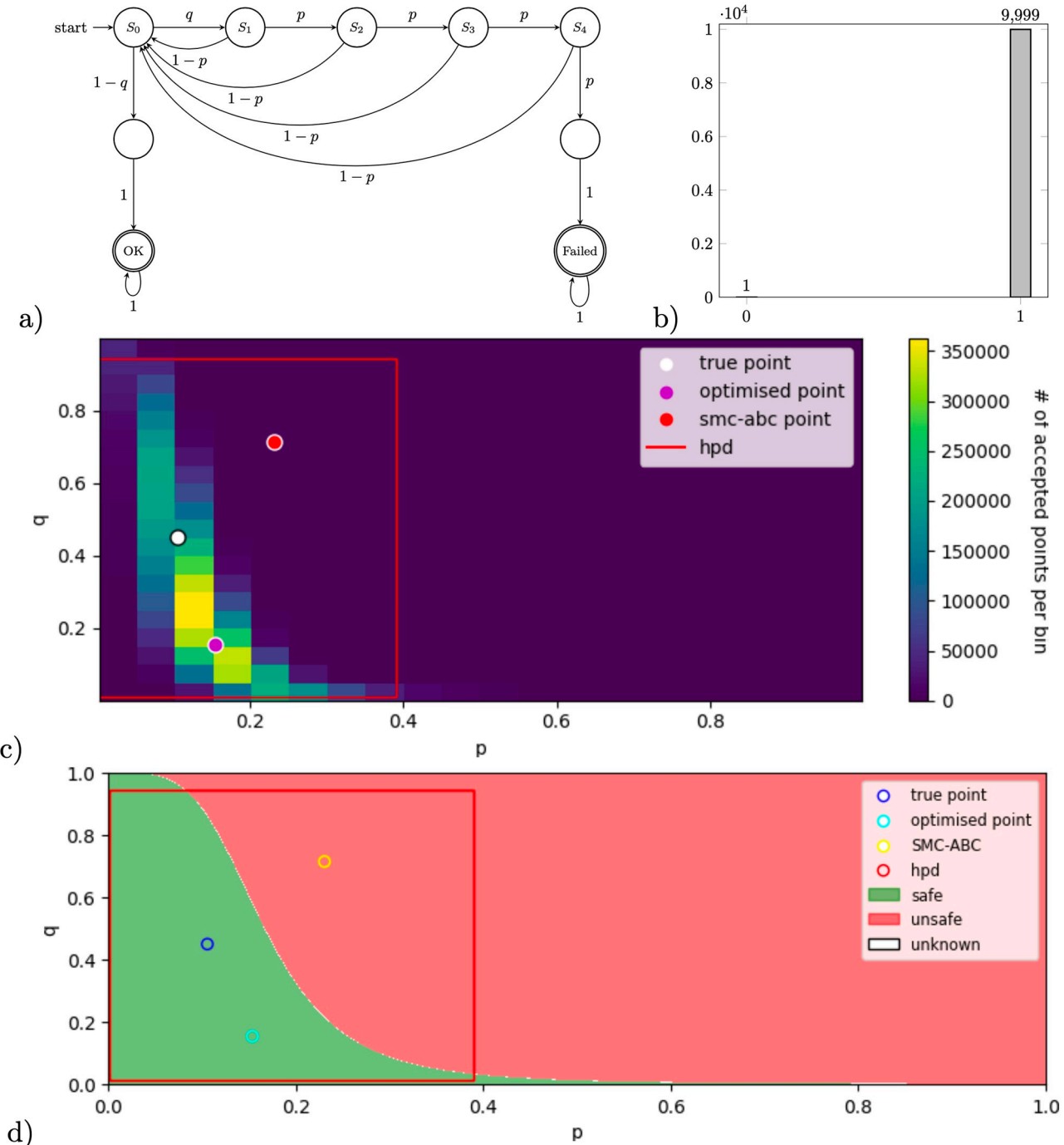

**Fig 7.** a) Zeroconf pMC for $n = 4$ probes. b) Data histogram of reaching the respective BSCCs as a result of 10,000 simulations using true parameter values. c) Visualisation of RF-MH results after 47,953,711 iterations, while trimming the first 25% of 10,086,444 accepted points obtained in 1 hour processor time using PC Skadi. The true point is shown as a white dot, the result of RF-opt shown as a purple dot, and the result of SMC-ABC shown as a red point. d) Visualisation of RF-ref results, where the green area (safe subspace) shows parameters which evaluate the rational functions within the intervals created from data using the corrected confidence level, 0.975. The true point is shown as a blue dot, the result of RF-opt shown as a cyan dot, and the result of SMC-ABC shown as a yellow point. In both figures, c) and d), the red rectangle shows the 95% HPD (highest posterior density) credible set, obtained by SMC-ABC.

reaching the second BSCC is a complementary event). Both results of RF-ref and RF-MH indicate a non-linear dependence of two parameters in the viable space of parameters. As in the previous example, RF-opt and SMC-ABC produce only single point estimates. While the optimised point of RF-opt is far from the true point but still in the possible parameter space, the point of SMC-ABC is not even in this space. The 95% HPD credible set computed by the SMC-ABC method provides on over-approximation of the viable parameter space computed by other methods, yet, as it is a hyper-rectangle in two dimensions, it does not capture the non-linear dependence seen by other methods.

We mathematically explain the dependency seen in the visualisations in Lemma 3.

**Lemma 3** (*parameters in Zeroconf example*) Assume a Zeroconf model with $n$ probes, such that the probabilities of reaching state 'OK' and 'Failed' are $\mu_{ok}$ and $\mu_f$ respectively. Then, the model parameters $p$ and $q$ are correlated according to a non-linear function that depends on two values: (i) $\alpha := \frac{\mu_{ok}}{\mu_f}$, the ratio of observations in 'OK' to 'Failed' (which determines the horizontal scaling in the green area shown in Fig 7d), and (ii) the number of probes $n$ (which determines the steepness of the function dividing green and red are as in Fig 7d).

See Fig 8 for a visual analysis of the influence of values $\alpha$ and $n$ on the correlation between $p$ and $q$. The Python script to produce the visualizations is available in the Zenodo package (`Zeroconf_correlation.ipynb`).

**Proof 2** *The long-run probability of ending up in BSCC OK is equal to*

$$
\begin{aligned}
\mu_{ok} &= \sum_{i=0}^{\infty} (\sum_{j=0}^{n-1} qp^j(1-p))^i \cdot (1-q) \\
&= \sum_{i=0}^{\infty} (\sum_{j=0}^{n-1} q(p^j - p^{j+1}))^i \cdot (1-q) \\
&= \sum_{i=0}^{\infty} (q \sum_{j=0}^{n-1} p^j - p^{j+1})^i \cdot (1-q) \\
&= \sum_{i=0}^{\infty} (q(1-p^n))^i \cdot (1-q) \\
&= \frac{1-q}{1-q+qp^n}
\end{aligned}
$$

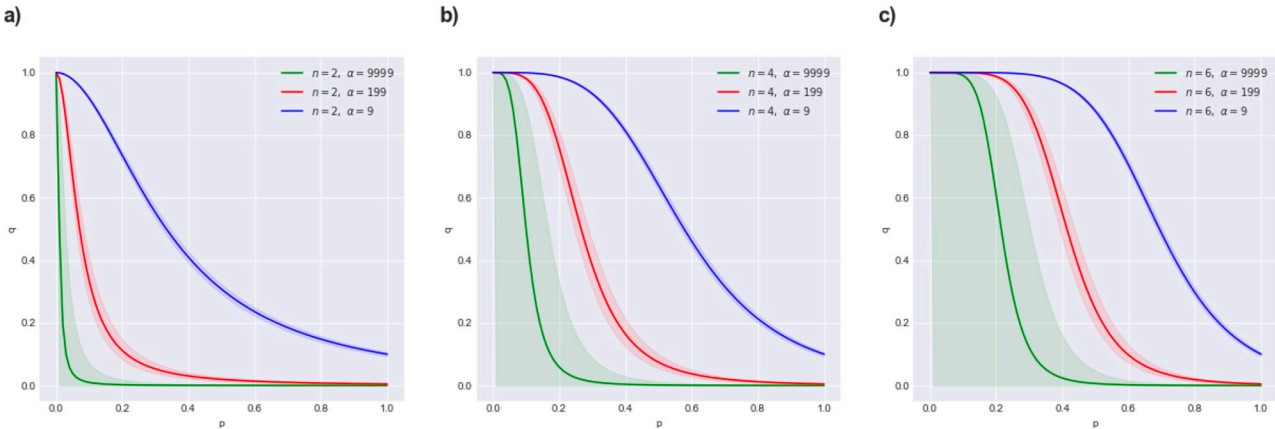

**Fig 8. Visualisation of the correlation between model parameters $p$ and $q$ in the Zeroconf example.** The correlation is based on a non-linear function which depends on two values: $\alpha$, the ratio of observations in 'OK' to 'Failed', and $n$, the number of probes. We varied $\alpha$ in each plot (green: $\alpha = 9999$, red: $\alpha = 199$, blue: $\alpha = 9$) and $n$ across plots (a: $n = 2$, b: $n = 4$, c: $n = 6$). The shaded areas represent the respective corrected 97.5% Agresti-Coull confidence intervals.

*Solving this equation for q results in $q = \frac{1-\mu_{ok}}{\mu_{ok}p^n-\mu_{ok}+1}$. Since the probability of ending up in the other BSCC Failed is $\mu_f = 1 - \mu_{ok}$, we can write $q = \frac{1}{\frac{\mu_{ok}}{\mu_f}p^n+1}$ . The function for q therefore describes the shape of the satisfaction area depending on p and is determined by two values, $\alpha = \frac{\mu_{ok}}{\mu_f}$ and n in the following way: (i) α describes the horizontal scaling—the function is closer to 0 for greater α (note this corresponds to more observations ending up in OK than Failed); accordingly, α also determines the 'endpoint', so the value for q at p = 1 equals $q = \frac{1}{\alpha+1}$. (ii) On the other hand, n describes the steepness of the function. Note that $p \leq 1$, and therefore q is close to 1 for small p and large n. Consequently, the shape of the function is more convex at lower values of p and drops to the endpoint of q at a greater value of p.*

## Conclusions and future work

In this paper, we investigated how the formal methods for parameter synthesis can aid parameter inference for parametric DTMCs when only steady-state data is available. Unlike directly running inference procedures that must approximate likelihood of data, we propose how to first use formal methods to obtain the *exact* likelihood for given data in terms of rational functions over parameters of the MC. Subsequently, we propose how these rational functions can be used to:

(i) efficiently compute the parameter points through maximising data likelihood (RF-opt method),

(ii) compute the viable space of parameters complying with the data in the sense of traditional interpretations of uncertainty at a desired confidence level (RF-alg and RF-ref method),

(iii) use rational functions to reduce uncertainty and boost scalability, through invoking them within a Bayesian MCMC parameter inference scheme.

The performance of these methods is compared with the likelihood-free Bayesian inference algorithm combining sequential Monte Carlo and approximate Bayesian computation (SMC-ABC). Results are reported over a motivating example with two parameters and four case studies:

(i) two ten-dimensional models—an artificial nested branching model and a real-world model representing honeybee mass-stinging where parameters are identifiable from steady-state data, but with challenging uncertainty propagation, and

(ii) two two-dimensional models—SIR epidemiological model and zero-configuration network protocol where parameters are not uniquely identifiable from steady-state data, and hence a more global analysis of viable parameter space is necessitated.

We demonstrated that our proposed method with first pre-computing the rational functions with formal methods brings a two-fold advantage. First, with the case studies of nested branching and honeybee defence attack, we demonstrate that our methodology allows one to significantly enhance the accuracy, precision and scalability of inference with respect to the typically employed sampling-based, likelihood-free techniques. Second, with the case study of SIR and Zeroconf protocol, we demonstrate that by inferring the *whole* viable parameter space (instead of only single value estimates), our methodology provides accurate, well-informed results of the correlation between parameters. Capturing the global viable parameter space in case of unidentifiable parameters becomes especially important for more complex models, where the shape of it quickly becomes non-trivial and unfeasible to derive mathematically by

hand. Our presented methods do not only compute the correlation between parameters automatically, but they also provide non-linear boundaries for the estimates.

A limitation of the proposed approach is that, for larger models (pMCs), the synthesis of rational functions can become infeasible due to memory management issues. Moreover, evaluating rational functions may become computationally expensive or subject to numerical errors. In these cases, the alternative is employing suitable model-abstraction techniques or resorting to statistical approximations which in turn necessitate instantiating the chain by statistical sampling. Population models induced from a counting abstraction, as in the honeybee collective defence model we studied here, have been widely studied in the context of biochemical reaction networks. Ideas focusing on the faster prediction of resulting distributions over subpopulations of molecular species, based on fluid, continuous space approximations [54, 55], as well as moment closure approximations [56–58] could be useful for improving the scalability of our parameter synthesis problem. Further promising approaches include global optimisation algorithms adopted from machine learning ideas, allowing us to develop a notion of robustness degree [59, 60]. Different from our work here, these approaches handle continuous-time Markov models and assume temporal data.

While the methodology presented in this paper trivially applies to the case when all BSCCs are singletons, it can be applied to the more general case when BSCCs contain more states but are indistinguishable by the observational apparatus. In future work, we plan to further generalise and evaluate our framework for more complex temporal properties and for the case when the BSCCs contain states with different labels. Moreover, we plan to investigate how different projections of data can help to reduce uncertainty in the inference procedures.

## Author Contributions

**Conceptualization:** David Šafránek, Tatjana Petrov.

**Data curation:** Julia Klein, Matej Hajnal.

**Formal analysis:** Julia Klein, Matej Hajnal, Tatjana Petrov.

**Investigation:** Julia Klein, David Šafránek, Tatjana Petrov.

**Methodology:** Julia Klein, Huy Phung, David Šafránek, Tatjana Petrov.

**Software:** Julia Klein, Huy Phung, Matej Hajnal.

**Supervision:** David Šafránek, Tatjana Petrov.

**Validation:** Matej Hajnal.

**Visualization:** Julia Klein, Matej Hajnal.

**Writing – original draft:** Julia Klein, David Šafránek, Tatjana Petrov.

**Writing – review & editing:** Julia Klein, Matej Hajnal, David Šafránek, Tatjana Petrov.

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
