## [Decision Letter · Decision Letter 0]

25 Jun 2023

PONE-D-23-14562Combining formal methods and Bayesian approach for inferring discrete-state stochastic models from steady-state dataPLOS ONE

Dear Dr. Petrov,

Thank you for submitting your manuscript to PLOS ONE. After careful consideration, we feel that it has merit but does not fully meet PLOS ONE’s publication criteria as it currently stands. Therefore, we invite you to submit a revised version of the manuscript that addresses the points raised during the review process.

We look forward to receiving your revised manuscript.

Kind regards,

Ruriko Yoshida

Academic Editor

PLOS ONE

Additional Editor Comments:

I agreed with the referee 2's comments on clarification. I think the authors need to clarify the manuscript for more general audience.

Reviewers' comments:

Reviewer's Responses to Questions

**Comments to the Author**

1. Is the manuscript technically sound, and do the data support the conclusions?

Reviewer #1: Yes

Reviewer #2: Yes

2. Has the statistical analysis been performed appropriately and rigorously? 

Reviewer #1: Yes

Reviewer #2: Yes

3. Have the authors made all data underlying the findings in their manuscript fully available?

Reviewer #1: Yes

Reviewer #2: Yes

4. Is the manuscript presented in an intelligible fashion and written in standard English?

Reviewer #1: No

Reviewer #2: Yes

5. Review Comments to the Author

Reviewer #1: The article is well-written and worthy of publication. I see no mathematical errors. The manuscript contains numerous grammatical, phrasing and spelling errors that need to be corrected prior to publication.

Reviewer #2: The authors present a framework for estimating the parameters of a discrete time Markov chain given only the label of the final state. The concepts are demonstrated through four simulation and case studies. The content is presented in a very rigorous framework and highly technical language.

Given that PLOS One targets general audience, I believe the manuscript needs a significant attempt to clarify the writing. I have the following specific suggestions. I suggest not using acronyms in the abstract (i.e., BSCC). Additionally, the concept of formal method needs to be defined or given an appropriate reference (line 60).

From my understanding, the estimation also relies on knowing the structure of the DTMC in order to write down all of the polynomials of the parameters. However, I did not understand from the current manuscript how this assumption comes into play. Please clarify the assumptions of the framework in the appropriate section. Additionally, how can a practitioner hypothesize the connections in order to use the estimation methods?

- Line 17: suggestion – don’t repeat the phrase after the acronym is introduced

- Suggest rephrasing lines 19-35 for clarity

- Lines 36-47: references? These are strong claims

- Line 84: What’s SMT?

- Line 101: typo? “elaborated in towards”

- Line 390: typo “with regarding their weights.”

- Line 453: missing word “ would allow to explore”

6. PLOS authors have the option to publish the peer review history of their article (what does this mean?). If published, this will include your full peer review and any attached files.

Reviewer #1: **Yes: **Daniel J. Hrozencik

Reviewer #2: No

---

## [Author Response · Author response to Decision Letter 0]

7 Aug 2023

The response to reviewers is previously uploaded in a separate file named "response_letter.pdf".

---

## [Decision Letter · Decision Letter 1]

23 Aug 2023

Combining formal methods and Bayesian approach for inferring discrete-state stochastic models from steady-state data

PONE-D-23-14562R1

Dear Dr. Petrov,

We’re pleased to inform you that your manuscript has been judged scientifically suitable for publication and will be formally accepted for publication once it meets all outstanding technical requirements.

Kind regards,

Ruriko Yoshida

Academic Editor

PLOS ONE

Additional Editor Comments (optional):

Reviewers' comments:

Reviewer's Responses to Questions

**Comments to the Author**

1. If the authors have adequately addressed your comments raised in a previous round of review and you feel that this manuscript is now acceptable for publication, you may indicate that here to bypass the “Comments to the Author” section, enter your conflict of interest statement in the “Confidential to Editor” section, and submit your "Accept" recommendation.

Reviewer #1: All comments have been addressed

Reviewer #2: (No Response)

2. Is the manuscript technically sound, and do the data support the conclusions?

Reviewer #1: Yes

Reviewer #2: Yes

3. Has the statistical analysis been performed appropriately and rigorously? 

Reviewer #1: Yes

Reviewer #2: Yes

4. Have the authors made all data underlying the findings in their manuscript fully available?

Reviewer #1: Yes

Reviewer #2: Yes

5. Is the manuscript presented in an intelligible fashion and written in standard English?

Reviewer #1: Yes

Reviewer #2: Yes

6. Review Comments to the Author

Reviewer #1: All my concerns have been adequately addressed. Thank you for your hard work on a well-written manuscript.

Reviewer #2: The authors have made substantial improvements to the readability of the manuscript. They have satisfactorily addressed previous reviewer concerns. In particular, the introduction of alternative phrases for previous jargon and expanded introduction improve the accessibility of the content for the target audience. I recommend acceptance after addressing the following minor comments and thoroughly proofreading.

Line 89: “and optimising…” rephrase; should the ‘and’ be deleted?

Line 661: lower case “More” after the semicolon

Line 665: remove the personal note to self (to do: rewrite `lossy`)

7. PLOS authors have the option to publish the peer review history of their article (what does this mean?). If published, this will include your full peer review and any attached files.

Reviewer #1: **Yes: **Daniel Hrozencik

Reviewer #2: **Yes: **Toryn Schafer

---

## [Editor Report · Acceptance letter]

2 Nov 2023

PONE-D-23-14562R1 

Combining formal methods and Bayesian approach for inferring discrete-state stochastic models from steady-state data 

Dear Dr. Petrov:

I'm pleased to inform you that your manuscript has been deemed suitable for publication in PLOS ONE. Congratulations! Your manuscript is now with our production department. 

Kind regards, 

on behalf of

Dr. Ruriko Yoshida 

Academic Editor

PLOS ONE